# An Unnatural Amino Acid-Regulated Growth Controller Based on Informational Disturbance

**DOI:** 10.3390/life11090920

**Published:** 2021-09-05

**Authors:** Yusuke Kato

**Affiliations:** Institute of Agrobiological Sciences, National Agriculture and Food Research Organization (NARO), Oowashi 1-2, Tsukuba 305-8634, Ibaraki, Japan; kato@affrc.go.jp; Tel.: +81-29-838-6059

**Keywords:** synthetic biology, genetic parts, kill-switch, growth control, unnatural amino acids, codon reassignment, pyrrolysine

## Abstract

We designed a novel growth controller regulated by feeding of an unnatural amino acid, *N^ε^*-benzyloxycarbonyl-l-lysine (ZK), using a specific incorporation system at a sense codon. This system is constructed by a pair of modified pyrrolisyl-tRNA synthetase (PylRS) and its cognate tRNA (tRNA^pyl^). Although ZK is non-toxic for normal organisms, the growth of *Escherichia coli* carrying the ZK incorporation system was inhibited in a ZK concentration-dependent manner without causing rapid bacterial death, presumably due to generation of non-functional or toxic proteins. The extent of growth inhibition strongly depended on the anticodon sequence of the tRNA^pyl^ gene. Taking advantage of the low selectivity of PylRS for tRNA^pyl^ anticodons, we experimentally determined the most effective anticodon sequence among all 64 nucleotide sequences in the anticodon region of tRNA^pyl^ gene. The results suggest that the ZK-regulated growth controller is a simple, target-specific, environmental noise-resistant and titratable system. This technique may be applicable to a wide variety of organisms because the growth inhibitory effects are caused by “informational disturbance”, in which the highly conserved system for transmission of information from DNA to proteins is perturbed.

## 1. Introduction

Growth control is a crucial technique for applications in microbial biotechnology [1]. Bacterial cells have a limited number of resources and there is a trade-off between microbial growth and all other cellular functions, including production of useful proteins or metabolites. This suggests that precise growth control is necessary to achieve maximum efficiency of microbial production. Therefore, there is ongoing development of methods to control growth. Classically, antibiotics, nutritional restriction, and temperature control have been used [2,3,4,5].

Multi-cell systems have many advantages over traditional clonal population systems, but each cell population must be regulated to optimize its percentage in the total population [6,7]. This challenge has prompted studies of more precise growth control for a specific population. A genetic controller that can regulate growth of specific subpopulations in a titratable and precise manner would achieve the ideal intermediate product supply-and-demand balance and optimize the efficiency of biological production. Several methods for growth control have been described, including synthetic growth switches based on controlled expression of RNA polymerase or methionine synthetase [8,9]. These methods are effective, but also have some inherent drawbacks, such as non-titratable regulation and use of dropout media [9,10]. Engineered cell–cell signaling systems have been proposed to enable autonomous coordination of subpopulation densities, but these sophisticated systems require construction of complex artificial genetic circuits [10,11,12,13,14].

We previously reported a tight and titratable translational controller for toxic protein production and biological containment (Figure 1a) [15,16,17]. This technique is based on conditional translation of target proteins using site-specific unnatural amino acid (Uaa) incorporation [15,18,19,20,21]. In this system, a Uaa-specific tRNA synthetase (UaaRS) and its cognate tRNA, which incorporates the Uaa at the UAG stop codon, are expressed in the cells of a target organism. In addition, a single or several UAG codons are inserted into the coding region of target genes. Translation of the UAG-inserted target genes is interrupted in the absence of Uaa, but functional proteins are produced in the presence of Uaa by UAG stop codon readthrough [18].

Here, we developed a simple growth control device based on a Uaa-regulated growth controller, which is conceptually similar to the Uaa-regulated translational controller. This growth controller incorporates the Uaa at a sense codon that is assigned to a normal proteinous amino acid (Figure 1b). The Uaa is competitively incorporated into many proteins against the originally assigned amino acid, resulting in production of abnormal proteins that may be non-functional or toxic (Figure 1c) [22]. Finally, growth of the target organism is inhibited by disturbance of a wide range of biological process.

The Uaa-regulated growth controller was designed to satisfy four conditions of an ideal growth control device. First, non-target organisms should not be affected. For use in the site-specific Uaa incorporation system, the Uaa must be chosen such that it is not recognized by host aaRSs and does not have inherent cytotoxicity because the Uaa’s are incorporated into proteins using the translation machinery in viable cells, suggesting that such Uaa’s are not harmful to non-target organisms [23]. Second, growth inhibition should be resistant to environmental noise. The Uaa must not be found in the natural environment and must not be one of the many non-proteinous amino acids present in cells as precursors to natural amino acids or other components of cellular metabolism, suggesting that accidental contamination from these sources will not cause unexpected growth inhibition. Thus, intentional feeding of a Uaa is the only way to inhibit the growth of the target organisms. Third, the device should be as simple as possible genetically. This is achieved because the Uaa-regulated growth controller includes only two genes: a Uaa-specific tRNA synthetase and its cognate tRNA. Any versatile genetic elements such as inducible promoters are not needed. Fourth, excellent future scalability is required. The codon-amino acid assignment is stably conserved, with some exceptions [24]. A limited assignment change is likely to alter expression of many genes in the cell and inhibit a wide range of biological process [25]. The harmful effects of the Uaa-regulated growth controller are caused by “informational disturbance”, through perturbance of the highly conserved system of transmission of information from DNA to proteins. Therefore, a Uaa-regulated growth controller should be effective in a wide range of organisms. In addition, several hundred site-specific Uaa incorporation systems have already been established that can be used to build growth control devices [26]. In particular, the fourth characteristic is a unique feature that is not found in conventional growth controllers.

As mentioned above, it is theoretically possible to create a device to control microbial growth with Uaa’s. However, it is uncertain if missense incorporation, in which a different amino acid is introduced into the sense codon than the originally assigned amino acid, slows down microbial growth or causes cell death. Several previous studies have shown that bacteria are broadly tolerant of proteome-wide missense substitutions with a normal proteinous amino acid [27,28,29,30]. In addition, ribosomal ambiguity (ram) mutants, in which ribosomal protein S4 is altered, also cause an increase in misreading, but grow at approximately the rate of the wild type [31]. On the other hand, it has been suggested that the efficacy of aminoglycoside antibiotics that target bacterial ribosomes is due to their ability to induce missense incorporation of amino acids [32,33,34,35]. Therefore, the concept of the Uaa-regulated growth controller needs to be proved experimentally. In this study, we examined the underlying principles of the Uaa-regulated growth controller.

## 2. Materials and Methods

### 2.1. Strains and Growth Conditions

BL21-AI[*F ompT gal dcm lon hsdSB(rB mB) araB*::*T7RNAP tetA*] was used throughout the study [36]. Bacteria were grown in Luria-Bertani (LB) medium at 37 °C. For preparation of solid medium, an aliquot of agar (2%) was added.

### 2.2. Plasmid Construction

The plasmid pTK2-1 ZLysRS1 that encodes the parent ZKRS and tRNA^pyl^, driven by the *trpS* promoter and *lpp* promoter, respectively, was provided by Kensaku Sakamoto and Shigeyuki Yokoyama (RIKEN) [37]. A plasmid library containing tRNA^pyl^ genes with randomized anticodon sequences was constructed as shown in Figure 2a. The anticodon sequence was randomized using inverse PCR with a random three-nucleotide sequence in the anticodon region and 16 bp overlapped primers, followed by circularization using enzymatic recombination (In-Fusion HD cloning kit, Takara, Kusatsu, Japan). PCR was performed using a high-fidelity DNA polymerase (KOD-plus-Neo, Toyobo, Osaka, Japan). Circularized PCR products were transfected into electrocompetent *E. coli* cells by electroporation using a Gene Pulser II electroporator (Bio-Rad, Hercules, CA, USA). A total of 191 strains were isolated and subjected to a preliminary ZK-sensitivity test. Plasmids were purified from these strains and the anticodon region of tRNA^pyl^ genes was sequenced. Anticodon sequences that were not found in the 191 clones were synthesized as described above, using a primer containing the missing anticodon sequence.

### 2.3. ZK-Sensitivity Test

A preliminary test (Figure 2b) was performed using a LB-agar plate. Tested strains carrying pTK2-1 ZLysRS1 variants containing various anticodon sequences were cultured in 1 mL of LB medium containing chloramphenicol (50 μg/mL). The overnight (about 16 h) culture of bacteria was 10^3^-fold diluted in fresh LB medium. An aliquot of diluted bacterial suspension (250 μL) was inoculated on LB-agar plates containing 3 mM ZK (Bachem). After an overnight culture, growth inhibition was evaluated by comparing bacterial growth on the ZK-containing plate with that on the ZK-free plate.

For a more precise test (Figure 2c and Figure 3a), a liquid culture was used. An overnight culture of tested strains was 10^3^-fold diluted in LB medium without chloramphenicol and incubated for 5 h. After the incubation, the bacterial culture was 10^2^-fold diluted in fresh LB medium containing ZK without chloramphenicol. The OD_590_ was measured at specified times and growth inhibition was evaluated by comparing the OD_590_ in ZK-containing medium with that of ZK-free medium.

To determine the mode of action, the time course of cell viability was measured (Figure 4). An overnight culture of tested strains was 10^4^-fold diluted in LB medium containing ZK with or without chloramphenicol. An aliquot (10 µL) was withdrawn at specified time points. After a 10^2^-fold dilution, the bacterial suspension was inoculated onto a ZK-free LB-agar plate. After an overnight culture, the number of colonies was counted to evaluate cell viability. As a preliminary test (Appendix A), the viability was measured only at times 0 and 6 h.

## 3. Results

The Uaa-regulated growth controller was constructed and characterized in *Escherichia coli.* Pyrrolysyl-tRNA synthetase (PylRS) of the archaeon *Methanosarcina mazei* aminoacylates a unique proteinous amino acid pyrrolysine onto its cognate tRNA (tRNA^pyl^) [38,39,40]. PylRS-tRNA^pyl^ is orthogonal to both prokaryotic and eukaryotic cells; i.e., PylRS does not aminoacylate any host tRNAs and host aminoacyl-tRNA synthetases (aaRSs) do not aminoacylate tRNA^pyl^, and pyrrolysine is not catalyzed by host aaRSs and no other canonical proteinous amino acids are catalyzed by PylRS [41]. Many modified PylRSs that specifically recognize Uaa’s have been generated to incorporate the Uaa into ribosomally synthesized proteins in various host cells [26]. We used a modified PylRS that specifically charges tRNA^pyl^ with the unnatural amino acid *N**^ε^*-benzyloxycarbonyl-l-lysine (ZK) to construct the Uaa-regulated growth controller [19,42]. The natural tRNA^pyl^ incorporates the Uaa’s at UAG because the anticodon sequence is CUA [42].

Fortunately, PylRS does not use the anticodon sequence of tRNA^pyl^ for substrate tRNA recognition, which suggests that the anticodon sequence can be modified to incorporate Uaa’s at other codons [41,43]. The anticodon sequence of the tRNApyl gene was randomized to incorporate ZK at various sense and nonsense codons to give 191 strains of E. coli BL21-AI carrying the plasmid containing the ZK-specific modified pylRS (ZKRS) gene and the tRNA^pyl^ gene with a randomized anticodon sequence (Figure 2a). In a preliminary test, the growth of some of these strains was markedly inhibited in the presence of ZK, although others were relatively resistant (Figure 2b).

The plasmid was isolated from each strain and the anticodon sequence of the modified tRNApyl gene was determined. The relationship between growth inhibition and ZK concentration was also determined to evaluate the ZK sensitivity precisely. The ZK sensitivity was clearly affected by the anticodon sequence (Figure 2c). A submilimolar concentration of ZK suppressed growth of some sensitive strains, and growth inhibition became more severe at a higher concentration. The concentration–inhibition curve was shifted to a higher concentration range in less-sensitive strains.

Anticodon sequences not found in the 191 strains were newly synthesized and transfected into bacterial cells to reveal the complete relationship between ZK sensitivity and anticodon sequence. Finally, a complete collection of anticodon sequences complementary to all 64 codons was prepared and the ZK sensitivity was determined (Figure 3a and Appendix A). The magnitude of growth inhibition was confirmed to be dependent on the anticodon sequence of the tRNA^pyl^ gene.

**Figure 3 life-11-00920-f003:**
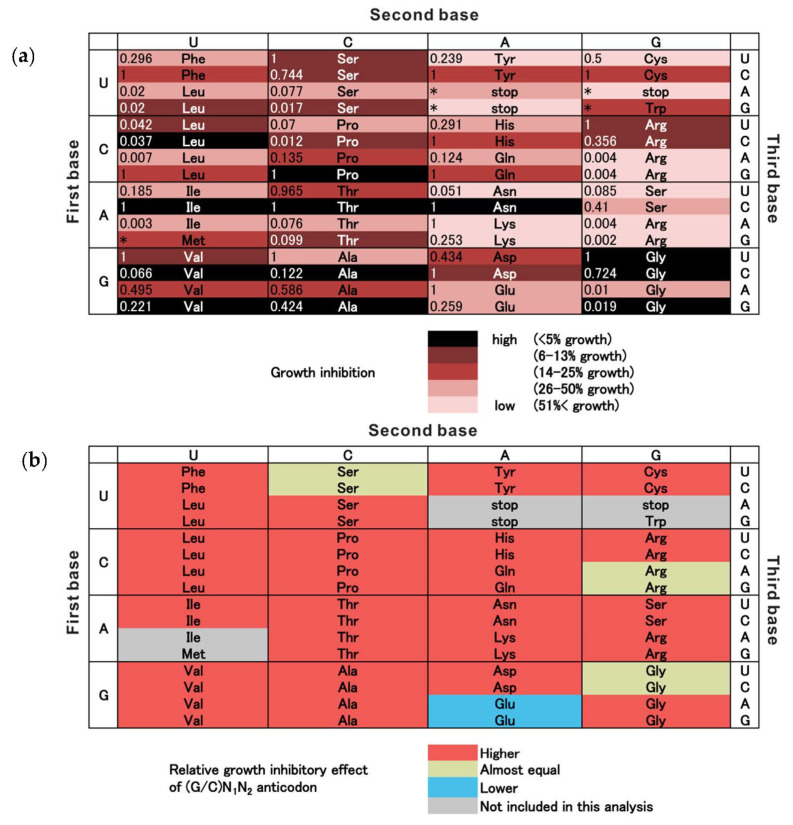
Anticodon sequence dependency of growth inhibition. (**a**) Relationship between anticodon sequence and growth inhibition. Inhibition of bacterial growth in the presence of 3 mM ZK was measured. Bacterial growth was evaluated relative to growth in the absence of ZK. The experiment was performed in the absence of chloramphenicol as a selection marker antibiotic because ZK incorporation at sense codons may also affect chloramphenicol sensitivity. For details of the experiment, see Materials and Methods and Appendix A. Anticodon sequences are shown as corresponding codon sequences. Note that the actual codons recognized by each anticodon cannot be determined definitively because the post-transcriptional modification of the tRNAs is not clear and there is a wobble effect. Amino acids assigned to the codons are also shown. The value to the left of the amino acid name shows the frequency of codon usage in highly expressed genes as the relative codon adaptiveness (RCA) [44,45]. RCA is defined as the ratio of the observed frequency of the target codon to the frequency of the most frequent synonymous codon, for which the RCA is rated as 1. (**b**) Higher growth inhibitory effects of GN_1_N_2_ and CN_1_N_2_ anticodons. Effects of GN_1_N_2_ anticodons, corresponding to (complementary N_2_N_1_)C codons, were compared with those of AN_1_N_2_, corresponding to (complementary N_2_N_1_)U codons (purine pairs). Similarly, effects of CN_1_N_2_ anticodons, corresponding to (complementary N_2_N_1_)G codons, were compared with those of UN_1_N_2_, corresponding to (complementary N_2_N_1_)A codons (pyrimidine pairs).

Several interesting correlations were observed in these results. Anticodons complementary to codons for small amino acids, such as Gly, Val, Ala and Pro, were frequently highly toxic. In contrast, those for larger amino acids, such as Arg, Lys, Trp, Tyr, and Phe, were mostly less toxic. However, these tendencies were not completely consistent and some exceptions were observed, such as the high toxicity of an anticodon corresponding to a codon for CGU(Arg) and CGC(Arg). Anticodons complementary to stop codons were weak or non-toxic. The relative growth was 1.01, 0.47, and 1.00 for UGA, UAA, and UAG stop codons, respectively.

It is noteworthy that the toxicity of tRNA^pyl^s containing GN_1_N_2_ and CN_1_N_2_ anticodons, corresponding to (complementary N_2_N_1_)C and G codons, were relatively higher than those of AN_1_N_2_ and TN_1_N_2_ anticodons, corresponding to (complementary N_2_N_1_)U and A codons, in many N_1_N_2_ pairs assigned to identical amino acids (Figure 3b and Appendix A). Of all the GN_1_N_2_-AN_1_N_2_ (purine) pairs and CN_1_N_2_-UN_1_N_2_ (pyrimidine) pairs included in the analysis, GN_1_N_2_ and CN_1_N_2_ showed higher growth inhibition in most pairs (25/29). In three pairs, there were no significant toxicity differences, and only one pair showed lower toxicity.

Having identified several strong growth inhibitory anticodon sequences, we next examined the mode of growth inhibition. The mode of antimicrobial action is usually categorized as bacteriostatic or bactericidal. Bacteriostatic antimicrobials suppress growth and proliferation, but bacteria do not die and growth is re-established after return to a permissive condition. In contrast, antimicrobials with a bactericidal action kill bacteria irreversibly. Preliminarily tests of the mode of action were performed against strains carrying tRNA^pyl^ genes with strong growth inhibitory anticodon sequences, corresponding to CUC(Leu), AUC(Ile), GUC(Val), GUG(Val), CCG(Pro), ACC(Thr), GCC(Ala), GCG(Ala), AAC(Asn), GGU(Gly), GGC(Gly) and GGG(Gly), based on measurement of the change in viability over time in the presence of 3 mM ZK and in the absence of chloramphenicol (Appendix A). Although some changes in the number of viable bacteria were observed, all tested strains remained viable after 6 h of incubation, suggesting that the mode of action may be bacteriostatic against most strains.

The detailed time courses of viability were determined for selected strains carrying tRNA^pyl^ genes containing anticodons CGC, GGT, and GGG, corresponding to GCG(Ala), ACC(Thr), and CCC(Pro), respectively. The CGC and GGT anticodons were strongly growth inhibitory and the GGG anticodon was intermediate (Figure 3a). The number of viable *E. coli* carrying a tRNA^pyl^ gene with a CGC anticodon sequence was well maintained in the presence of 3 mM ZK for 6 h, suggesting that bacteria survived without dying off quickly, despite severe growth inhibition (Figure 4a). For the GGT anticodon, the number of surviving bacteria was also well maintained until 4 h, but gradually decreased in 4 to 6 h (Figure 4b). These results confirm the bacteriostatic effect of the Uaa-regulated growth controller, although prolonged ZK treatment sometimes gradually killed the bacteria. As predicted from the experiments in Figure 2c, the growth rate for the GGT anticodon decreased, but the number of bacteria still increased at a low ZK concentration (0.3 mM). Similarly, with the CCC anticodon, only partial inhibition of bacterial growth was observed even in the presence of 3 mM ZK, suggesting that diverse degrees of growth inhibition can be obtained by the choice of anticodon or ZK concentration.

**Figure 4 life-11-00920-f004:**
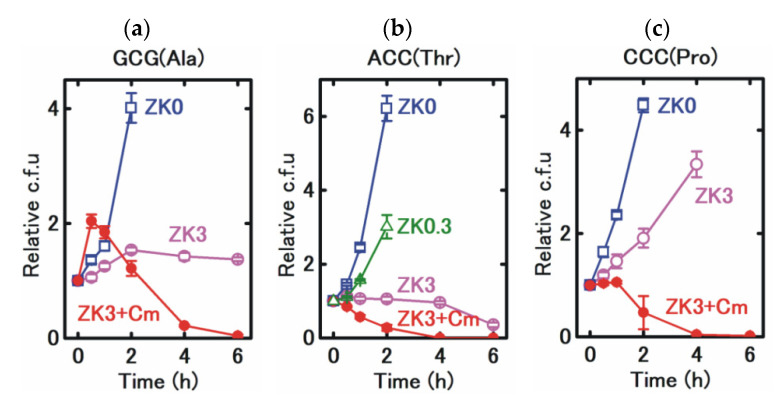
Mode of action of the growth controller. Time courses of viability are shown. The tested bacterial strains were incubated in the presence of 0 mM (ZK0), 0.3 mM (ZK0.3) or 3 mM ZK (ZK3), for the indicated time. The number of viable bacteria at each time point is represented as a relative value to the number of viable bacteria at time 0. Tests were performed for strains carrying a tRNA^pyl^ gene containing the CGC (**a**), GGU (**b**) and GGG (**c**) anticodon, complementary to the codons of GCG(Ala), ACC(Thr) and CCC(Pro), respectively. Cm, chloramphenicol. For experiments cultured in the presence of 3 mM ZK and in the absence of chloramphenicol, colonies derived from samples after 6 h of incubation were randomly selected and examined for plasmid maintenance using chloramphenicol resistance as an indicator. For all strains, most of the colonies maintained the plasmid (64/64 for all strains). Data points represent the mean ± s.d. of three biological replicates.

Interestingly, clearly distinct results were obtained in the presence of chloramphenicol. In all three strains tested, the viable count was markedly reduced. After 6 h of incubation in medium containing 3 mM ZK, only 4.1% of the viable count at time 0 was detected for the anticodon CGC, <0.2% for the anticodon GGT, and 1.1% for the anticodon GGG, suggesting that high concentrations of ZK have a bactericidal effect in the presence of chloramphenicol.

## 4. Discussion

A ZK-regulated growth controller in *E. coli* using ZKRS and tRNA^pyl^ with an anticodon corresponding to a sense codon was constructed and characterized in this study. Growth of bacteria carrying this growth controller containing the tRNA^pyl^ gene with an appropriate anticodon sequence was inhibited in the presence of ZK. The mode of action was mostly bacteriostatic. Our results show that bacterial growth can be suppressed to almost zero without killing bacteria by choosing the appropriate ZK concentration and anticodon sequence. In addition, growth inhibition could be titrated based on the ZK concentration. In contrast, ZK did not affect growth of bacteria carrying tRNA^pyl^ with non-toxic anticodons such as UGA and UAG stop codons, which indicates that ZK did not have any harmful pharmacological effects. These results suggest that the ZK-regulated growth controller is a simple, target-specific and titratable growth control system.

Growth inhibition was strongly dependent on the anticodon sequence of the tRNA^pyl^ gene. There may be several factors underlying this anticodon sequence dependency, but the precise causes were not examined. Theoretically, the inhibition mechanism can be divided into the following steps: efficient charging of tRNA^pyl^ with ZK, efficient ZK incorporation from ZK-tRNA^pyl^ into proteins, and growth inhibitory effects exerted by the ZK-containing proteins. All steps should be satisfied for strong growth inhibitory anticodons. In contrast, one or more of these steps may not be efficient for less toxic or non-toxic anticodons.

The first step, efficient charging of tRNA^pyl^ with ZK, may be influenced by aminoacylation by a host aminoacyl-tRNA synthetase, which uses the anticodon sequence for substrate tRNA recognition. For instance, in *Mycoplasma capricolum*, tRNA^pyl^_CCG_ is recognized well by the endogenous arginyl-tRNA synthetase and CGG codons are translated as Arg [46]. This type of mechanism may also interfere with ZK charging of tRNA^pyl^ containing other anticodon sequences because anticodons in tRNAs are typically used as recognition elements of most aaRSs in *E. coli* [47]. ZK incorporation into ZKRS should also be taken into account because the efficiency and substrate specificity may be affected.

Post-transcriptional nucleotide modification may affect the second step, efficient ZK incorporation from ZK-tRNA^pyl^ into proteins, but modification of *M. mazei* tRNA^pyl^ has not been widely studied in *E. coli* [48]. Modification of nucleotides is critical for all core aspects of tRNA function, such as folding, stability, and decoding [49]. The hypermodified base *N*^6^-threonylcarbamoyladenosine and its derivatives, which locate in the anticodon stem-loop at position 37 adjacent to the anticodon, is present in nearly all tRNAs that decode ANN codons [50,51]. This nucleotide modification stabilizes the anticodon loop, which promotes accurate decoding of ANN codons during protein synthesis [52,53]. The *N^6^*-threonylcarbamoyladenosine modification was not found in *Methanosarcina barkeri* tRNA^pyl^, suggesting that tRNA^pyl^_NNU_ is not optimized for decoding of ANN codons [54]. In addition, nucleotide modification at position 34, the first position of the anticodon in tRNA, modulates codon recognition, thereby promoting accurate decoding during protein synthesis. Thus, the nucleotide modification may complicate the relationship between anticodon sequences and growth inhibition [55].

The growth inhibitory effect of tRNA^pyl^ containing G or C at position 34 was higher than that containing A or U, respectively, in the box assigned to an identical amino acid. Previous reports suggest that tRNA^pyl^ is not optimized to incorporate amino acids in prokaryotes and eukaryotes [56,57]. Formation of a strong hydrogen bond between G and C at position 34, the first nucleotide of the anticodon and the third nucleotide of the codon, may contribute to efficient incorporation of ZK and partially optimize tRNA^pyl^ in the translation machinery of bacteria. Wobble recognition is another possible mechanism. For example, the GN_1_N_2_ anticodon is expected to decode the (complementary N_2_N_1_) U codon by wobble recognition, in addition to decoding the (complementary N_2_N_1_) C codon by Watson–Crick base pairing. The expansion of codons that can be decoded by wobble recognition will increase the site of introduction of ZK and thus make the anticodon more toxic. However, as mentioned above, the modification of tRNA^pyl^ bases, which greatly affects wobble recognition, is largely unknown in *E. coli*, so it is difficult at present to assess the effect of wobble recognition accurately [58].

We predicted that the second step was hampered by competitive incorporation of the natural proteinous amino acid that was originally assigned to the identical sense codon and charged to the host tRNA. Endogenous tRNA numbers were moderately inversely correlated with the sense codon reassignment efficiency in *E. coli* carrying a sense codon targeting a Tyr incorporation system using the *Methanocaldococcus jannaschii* TyrRS-tRNA^tyr^ [28]. In *E. coli*, the frequency of codon usage in highly expressed genes is directly proportional to the corresponding tRNA population [59,60]. However, contrary to expectations, tRNA^pyl^s corresponding to frequently appearing codons were often more growth inhibitory, including for codons assigned to Phe, Ile, Tyr, His, Gln, Asn, Asp, Cys, and Arg. This suggests that the competition between tRNA^pyl^ and endogenous tRNA is not a dominant determinant of toxicity (Figure 3a). A possible mechanism to explain this observation is that higher codon usage increases the number of sites for ZK incorporation, resulting in higher toxicity, and vice versa. Indeed, anticodon sequences complementary to some underutilized codons, such as CUA(Leu), AUA(Ile), CGA(Arg), CGG(Arg), AGA(Arg), and AGG(Arg), showed apparently low toxicity. However, some others showed an inverse correlation between codon usage and toxicity (e.g., Val, Ala).

It is reasonable to expect that substitutions of amino acids with other amino acids that are physicochemically distant will be more toxic because these substitutions have a higher probability of destroying the original conformation of a protein [61]. Therefore, the third step, a growth inhibitory effect exerted by ZK-containing proteins, is predicted to depend on which amino acid is substituted with ZK. This may explain the observation that anticodons corresponding to codons assigned to amino acids that are physicochemically distant from the bulky ZK, such as Gly, Val, Ala, and Pro, were highly toxic, whereas bulky amino acids that are similar to ZK, such as Arg, Lys, Trp, Tyr, and Phe, were less toxic or non-toxic. ZK incorporation at stop codons mostly results only in production of longer polypeptide chains. Moreover, the proteins in which Uaa’s were incorporated at endogenous stop codons were detected only at low levels [62]. These observations agree well with the result that anticodons corresponding to stop codons were less toxic or non-toxic. However, species-specific toxicity of stop codon suppression still needs to be considered carefully [63].

Overall, the relationship between growth inhibitory effect and anticodon sequence is difficult to predict because many factors may interact and affect the relationship, as described above. An advantage of our system is that strong growth inhibitory anticodon sequences can be identified experimentally in each target organism because of the low selectivity toward the tRNA^pyl^ anticodon of PylRS, which can be used as an orthogonal enzyme in both prokaryotes and eukaryotes [26]. In addition, incorporation of Uaa’s with physicochemical properties that are distinct from those of ZK may lead to a different spectrum of toxicity. Genetic incorporation of many Uaa’s has been reported using the PylRS-tRNA^pyl^-based system [41]. Therefore, the Uaa-regulated growth controller using a modified PylRS-tRNA^pyl^ is an excellent flexible platform to select the most effective Uaa and anticodon sequence for a target organism. Moreover, it will be possible to control multiple populations using several modified PylRS-tRNA^pyl^ pairs that recognize distinct Uaa’s orthogonally.

A high concentration of ZK showed a bactericidal effect in the presence of chloramphenicol. This observation is interesting because the bactericidal effect of the aminoglycoside antibiotic streptomycin, which induces missense incorporation, has been reported to be inhibited by chloramphenicol [64,65]. One possible mechanism of our observation is that the total activity of the selection marker chloramphenicol acetyltransferase (CAT) is severely reduced due to incorporation of ZK, resulting in impaired chloramphenicol resistance. From another perspective, this procedure can be used to eliminate specific sub-populations from microbial consortia. In the case of tRNA^pyl^_CGC_, the bacteria initially grew and then started to die. This delayed toxic effect can be explained by a time lag in adequate uptake of ZK into bacteria or in accumulation of ZK-incorporated proteins [17].

The disadvantage of this system is that it inhibits a wide range of protein synthesis, including that derived from transgenes, so it is difficult to control which cellular pathways are inhibited. This undesirable effect could be reduced by removing the target codon from genes of interest by synonymous codon substitution.

The site-specific Uaa incorporation system can be controlled not only by a Uaa, but also by expression control of UaaRS and/or its cognate tRNA [15,16]. This control of UaaRS and tRNA gene expression may be useful in connection of the Uaa-controlling kill-switch to a gene circuit.

## 5. Conclusions

The ZK-regulated growth controller is a simple, target-specific, environmental noise-resistant and titratable growth control device. The PylRS-tRNA^pyl^-based Uaa incorporation system is an ideal platform to construct an effective Uaa-regulated growth controller because this system allows experimental identification of the most effective anticodon sequence and Uaa. This system is also applicable in various organisms because PylRS-tRNA^pyl^ can be used in both prokaryotes and eukaryotes and the basis of growth inhibition is “informational disturbance”.

## Figures and Tables

**Figure 1 life-11-00920-f001:**
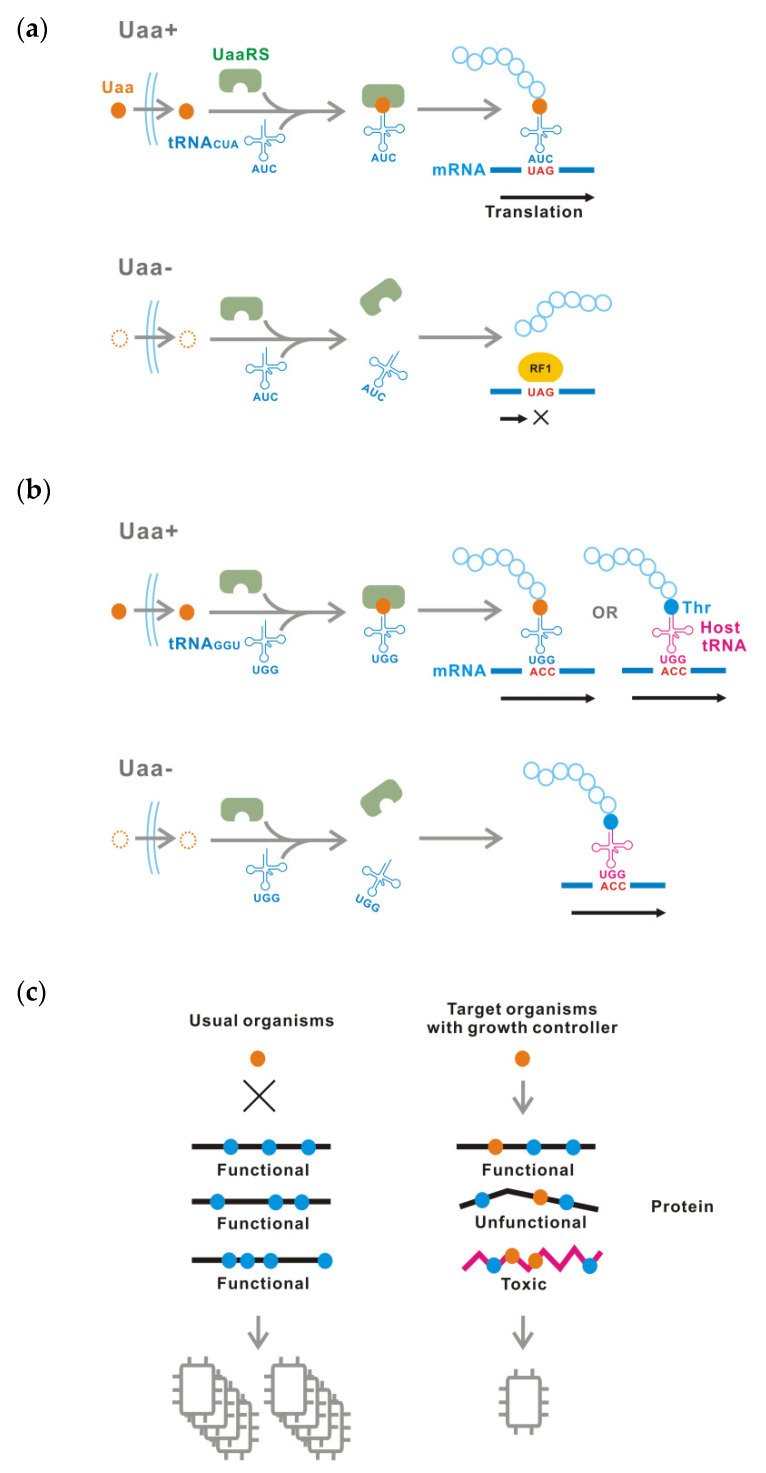
Principles of the Uaa-regulated growth controller. (**a**) Schematic of a Uaa-controlled translational switch using conditional UAG stop codon readthrough to construct synthetic auxotrophy for Uaa. (**b**,**c**) Schematic of a Uaa-regulated growth controller using Uaa incorporation at a sense codon. Uaa incorporation at the ACC codon assigned to Thr is shown as an example. In (**c**), bacteria are shown schematically as boxes.

**Figure 2 life-11-00920-f002:**
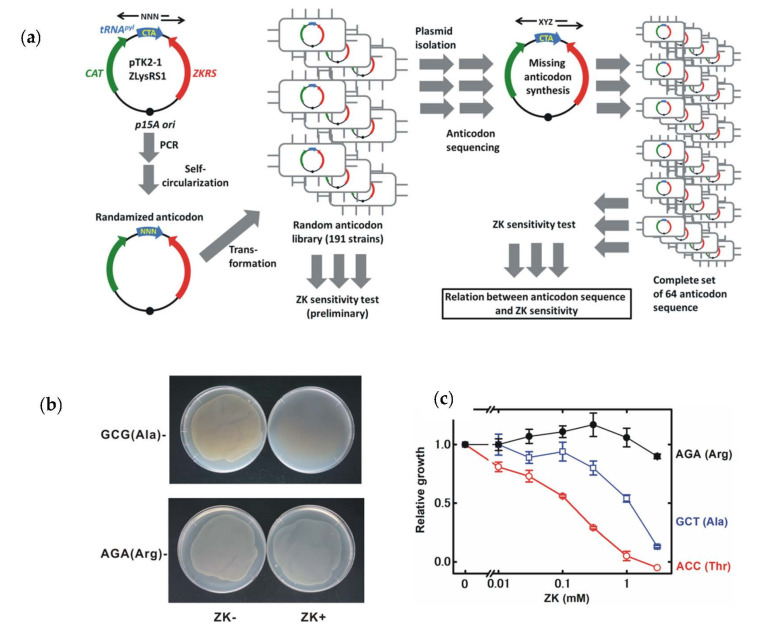
Growth inhibition of *E. coli* cells by activation of the ZK-regulated growth controller. (**a**) Construction of a tRNA^pyl^ gene library containing all 64 anticodons. Black arrows indicate PCR primers. N, non-specific base (A, T, G, or C). X–Z, specific bases defined in the study. (**b**) ZK-induced growth inhibition. *E. coli* carrying the ZK-regulated growth controller containing the tRNA^pyl^ gene with CGC and UCU anticodons, complementary to the GCG(Ala) and AGA(Arg) codons, respectively, was cultured with or without 3 mM ZK. (**c**) Relationship between ZK dose and bacterial growth inhibition. After 3 h of liquid culture in antibiotic-free medium, growth was evaluated based on OD_590_ values. Growth of bacteria was normalized to that in the absence of ZK. Data points represent the mean ± s.d. of three biological replicates. Very sensitive, moderately sensitive and resistant strains are shown. The complementary sequence of the anticodon and the assigned amino acid are indicated to the right.

## Data Availability

The data presented in this study are available on request from the corresponding author.

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
