# Peer review of "An Unnatural Amino Acid-Regulated Growth Controller Based on Informational Disturbance"

_life, 2021, doi:10.3390/life11090920_

Round 1
Reviewer 1 Report
Overall the idea of using directed missense incorporation of ncAAs as a cell type specific growth controllers in the context of designed consortia for biotechnological purposes is intriguing and potentially very useful. The initial results described suggest that this method may be useful and that the broad range of ncAAs, codons and aaRS available offer a wide space for tuning properties of such systems.
The results presented seem very preliminary and are not presented in a manner that makes it possible to clearly evaluate the proposed system. Description of experiments performed and the actual system under study were lacking. None of the figures showed error bars. The experimental descriptions were largely missing and when provided, were not detailed (lines 112-124 seem to describe all of the experiments performed?). Several of the broad conclusions drawn from the experiments are not backed up by clear evidence from experiments (statements outlined below are over interpretations and overly broad generalizations). Overall, this paper presents an interesting use of ncAA incorporating systems but requires extensive editing and inclusion of more quantitative experimental descriptions and data before it is ready for publication.
Page 3 starting line 66
First – modify the ncAA must be chosen such that it is not recognized by natural aaRSs and does not have inherent cytotoxicity
Second – The ncAA must not be one of the many non-proteogenic amino acids present in cells as precursors to natural amino acids or other components of cellular metabolism
The fourth point presents the theoretical picture of information transfer being easily disrupted at the protein level that is not backed up by the experimental evidence. High levels of missense incorporation are well tolerated by bacterial cells. Some discussion (and references) should be added to balance the argument against the general tolerance of bacterial cell to the presence of missense errors. Some reference to the broader the literature on the proximate and ultimate cytotoxic effects of missense inducing antibiotics might be warranted here.
A major concern is that the materials and methods need much more specific descriptions of the construct employed in the experiments. What exactly is the vector used, origin, promoters driving production of tRNA and aaRS? How has this system been benchmarked? What is the specific sequence of the orthogonal aaRS that was used? How has this sequence been characterized? What efficiency does it have? Does the observed cytotoxic effect depend on the identity of the introduced amino acid? All of these questions are related to the actual effect on cells that is being induced and the extent to which this effect is generalizable.
A major problem with the manuscript as written is that the many important quantitative details of the experiments and results are not adequately described. A specific description of what experiment was performed in order to generate the data in figure 3a is needed. Some of the quantitative information about these experiments (colony counts, replicate numbers std deviations…should be provided). A more specific description of the source and meaning of the codon usage number in the figure 3a is also needed. It is unclear to the reviewer why the numbers EITHER do not add up to one for each amino acid OR are capped at 1. Either they should be amino acid specific (all numbers for a particular amino acid need to add to 1) or should represent the overall amino acid usage in whatever the “highly expressed” protein fraction is (less commonly used amino acids would add to less than 1, more commonly used amino acids would add to more than 1).
The actual measured growth inhibitory values (as %) employing each specific anticodon sequence should be included in the figure or a related table.
Figure 3b seems an unnecessary duplication of figure 3a. Is there something in 3b that is not in 3a?
More description of the experimental details for the data presented in figure 2c is needed. How was the growth measured, in liquid on plates? Under what conditions (w/ w/o antibiotic)? How many biological replicates? Error bars on the data points are needed.
Line 187 “It is noteworthy that the toxicity of tRNApyls containing GN1N2 and CN1N2 antico-187 dons, corresponding to (complementary N2N1)C and G codons, were relatively higher than those of AN1N2 and TN1N2 anticodons, corresponding to (complementary N2N1)U 189 and A codons, in many N1N2 pairs assigned to identical amino acids (Fig 3b).”
How is this conclusion reached from your data? What specific metric is being applied to guage G/CNN vs A/TNN are different? How large of an effect is observed between different wobble reading tRNAs. Is the effect related to codon usage?
Line 204 “Preliminarily tests of the mode of action were performed against strains carrying tRNApyl genes with strong growth inhibitory anticodon sequences”
Provide a detailed experimental description and data for these experiments. Were the tested cells growing or stationary to start with? Were growth rates measured? What were growth rates in the presence and absence of ncAA?
Page 7 line 224-225 “These results suggest that the Uaa-regulated growth controller caused a bactericidal effect in the presence of chloramphenicol by attenuating the total activity of CAT.”
There is clearly a synergistic between the ncAA and antibiotic but there is no evidence that this effect is due to CAT deactivation. A clear experiment to demonstrate CAT involvement would be the production of a CAT gene that does not have any of the targeted codons. Absent this, or a similar experiment, the statement indicating CAT activity should be removed or more carefully made as a conjecture.
Lines 234 “Our results show that bacterial growth can be perfectly suppressed without killing bacteria by choosing the appropriate Uaa incorporation system, Uaa concentration and anticodon sequence. In addition, growth inhibition could be titrated based on the ZK concentration.”
The statement speaks to generalizability that was not measured in this study. The increased cytotoxicity approaching 6 hours is shorter than other bacteriostatic antibiotics. What would “perfect” control actually be? The effects of different ncAAs (or utility in different) strains were not measured. The titration control experiment (essentially the mode of action experiment with different concentrations of ZK) was not described. This statement should be amended to either describe the results presented or moved to a conclusion as a suggested possibilities of the approach.
Lines 261 “Modification of nucleotides is critical for all core aspects of tRNA function, such as folding, stability, and decoding [44].”
An alternative view is that the vast majority of tRNA modifications are dispensable with at most minor effects on cell growth. (only 4 of 35 E coli tRNA modifications are essential…. Ile tRNA lysidine, Arg tRNA A to I, proline tRNA m1G37 and Lys t6A37).
Lines 286 “However, contrary to expectations, tRNApyls corresponding to frequently appearing codons were often more growth inhibitory, including for codons assigned to Phe, Ile, Tyr, His, Gln, Asn, Asp, Cys and Arg.”
The result that more frequently USED codons seem to produce a more cytotoxic effect suggests that the toxicity may be correlated with the extent of proteome wide incorporation.
“It is reasonable to expect that substitutions of amino acids with other amino acids that are clearly physiochemically distant are highly toxic because these substitutions are likely to destroy the original conformation of a protein [55].”
The idea that mutations are disruptive of function is not correct (Kurland Ann Rev Gen 1992). That a single mutation in a given protein disables its function is not backed up by the effects of the majority of evaluated mutations. Typically surface mutations are completely neutral. If a substitution is destabilizing to a protein fold it is usually not inactivating. That is the average destabilization is less than the average stability of a protein fold. An important additional component of this analysis is that the insertion of an ncAA will only occur in a subset of translated proteins, such that many copies of unmodified proteins will be produced. The question of toxicity becomes two fold. Is the correctly formed (and % function of mutated) protein sufficient for the cells needs? Is the ncAA mutated protein in and of itself toxic? Does the cellular energy burn in creating partially functional protein also contribute to the effect?
Lines 324 “This delayed toxic effect can be explained by a time lag in adequate uptake of ZK into the bacteria [17]”
There is not evidence for this statement. A more probable interpretation is that the cytotoxic effect requires build up of modified protein via translation that takes time.
Reviewer 2 Report
In this manuscript the author reports regulation of E. coli growth through incorporation of unnatural amino acids into the nascent peptide. The findings here are potentially highly interesting to the field and that is why I feel that the writing and the presentation of the study should be improved.
The experiments are all performed in BL21 E. coli, which is genetically manipulated and commonly used for expression of recombinant proteins. Does the fact that these strains are deficient in Lon and OmpT proteases have any significance in the outcome of the experiments? This should be mentioned and discussed.
What is the reason behind choosing this background rather than the wild-type K-12 derivative lab strains such as MG1655 or BW25113?
How is the transcription of the Uaa tRNA synthetase and the tRNA, coded on the pTK2-1, regulated? Are they constitutively expressed?
In general, the manuscript should be more streamlined. The background and in particular the previous work done on growth control using unnatural amino acids should be discussed better in the Introduction and repeating should be avoided.
The first paragraph of the Results sections is not results and should be moved to Introduction.
Importantly, the aim of the study and its implications should be clearly stated and left to the reader to figure out. What is the use of having a titratable growth control system? This should be made clear very early on to the reader.
The discussion is too long and has a lot of background information that should be moved to the Introduction (i.e. lines 282-290 and 312-317).
Minor points,
Lines 30-31 – what is the advantage of controlling growth with unnatural amino acids over the classical methods mentioned?
Line 50 – please add reference.
Line 136 – please remove bacteria.
Lines 163-165 – this conclusion cannot be made from a lawn of cells on a plate. General inhibition growth with addition on ZK is also a likely possibility. A control growth curve should be shown with wt cells with/without ZK to rule this out.
Lines 168-169 – Fig 2c, is the graded toxicity of ZN dependent of codon usage?
Line 194 – “Mode of action” of what? Please have a more informative title for the figure.
Figure 1 – bottom part of panel c, please write that the boxes represent live cells.
Figure 4 – commonly the CFU/ml are plotted. Here they adjusted, please specify to what and why. Also, if there is no selection for the plasmids that carry a toxic element to the cell, they are readily lost. This would also explain the higher toxicity with the added pressure of keeping the plasmids in the presence of Cm. This should be discussed.
Reviewer 3 Report
The author devised a system that dials the growth depending on the supplementation of unnatural amino acid. The author expressed both PylRS and tRNApyl in the cell and observed this system suppresses the growth upon adding a substrate unnatural amino acid. The author mutated the anticodon of tRNApyl to examine the effect of anticodon sequence and demonstrated that degree of growth suppression differs between tRNApyl bearing different anticodons.
It is interesting to see such variations in growth suppression between different anticodon sequences in tRNApyl, because susceptibility of codons to perturbation by competing Pyl-tRNApyl differs.
The manuscript is generally well written, and the results are interesting for readers of Life.
However, there are substantial concerns in data interpretation.
Questions
Major points
- The author should be careful about the interpretation regarding the interaction between anticodons and codons. The author assumes that anticodons only bind to codons with Watson-Crick base pairing. However, there are a variety of wobble interactions besides canonical Watson Crick base pairing, such as G-U wobble pairing, I-G, I-A, I-U interactions. For example, in the L187, the G starting anticodon decodes C starting codon and U starting codon via the wobble interaction. Similarly, U starting anticodon interacts with G and sometimes also U and C. The author should take into account these codon-anticodon interactions for the interpretation of the data. References such as doi:10.1128/ecosalplus.ESP-0007-2013 would be helpful.
- Based on this concern, the author should have another representation of the results instead of Fig 3a because the codon-anticodon binding is not 1-to-1 correspondence. The author should include the data of each anticodon in the table rather than connect those numbers to codons
- A Cm resistant experiment does not correctly address the transgene expression because different strains without Cm resistant cassette can have different susceptibility to Cm. The author should conduct control experiments using the cells that do not have CAT cassettes or directly measure the CAT activity.
Minor points
- In the title, "informational disturbance" is unclear about what this phrase means. "perturbation of genetic code deciphering" or something more specific phrase would be suitable.
- Since this system broadly disturbs protein synthesis and is highly uncontrollable regarding what cellular pathways are perturbated. The author should mention this limitation of this application.
- Is there a correlation between codon usage and toxicity by ZK system? For example, are highly used codons more toxic when it is decoded by tRNA-pyl?
Round 2
Reviewer 1 Report
The authors mostly addressed my concerns
Reviewer 3 Report
In the revised manuscript, the author fully address this reviewer's concerns.